# Forming a three-dimensional porous organic network via solid-state explosion of organic single crystals

Seo-Yoon Bae[1], Dongwook Kim[2], Dongbin Shin[3], Javeed Mahmood [1], In-Yup Jeon[1], Sun-Min Jung [1], Sun-Hee Shin[1], Seok-Jin Kim[1], Noejung Park [3], Myoung Soo Lah[2] & Jong-Beom Baek[1]

Solid-state reaction of organic molecules holds a considerable advantage over liquid-phase processes in the manufacturing industry. However, the research progress in exploring this benefit is largely staggering, which leaves few liquid-phase systems to work with. Here, we show a synthetic protocol for the formation of a three-dimensional porous organic network via solid-state explosion of organic single crystals. The explosive reaction is realized by the Bergman reaction (cycloaromatization) of three enediyne groups on 2,3,6,7,14,15-hexaethynyl-9,10-dihydro-9,10-[1,2]benzenoanthracene. The origin of the explosion is systematically studied using single-crystal X-ray diffraction and differential scanning calorimetry, along with high-speed camera and density functional theory calculations. The results suggest that the solid-state explosion is triggered by an abrupt change in lattice energy induced by release of primer molecules in the 2,3,6,7,14,15-hexaethynyl-9,10-dihydro-9,10-[1,2]benze-noanthracene crystal lattice.

[1] School of Energy and Chemical Engineering/Center for Dimension Controllable Organic Frameworks, Ulsan National Institute of Science and Technology (UNIST), 50 UNIST, Ulsan 44919, South Korea. [2] Department of Chemistry, Ulsan National Institute of Science and Technology (UNIST), 50 UNIST, Ulsan 44919, South Korea. [3] Department of Physics, Ulsan National Institute of Science and Technology (UNIST), 50 UNIST, Ulsan 44919, South Korea. Correspondence and requests for materials should be addressed to J.-B.B. (email: jbbaek@unist.ac.kr)

Solid-state reaction of organic molecules has attracted considerable interest due to its environmental advantages and sustainability[1]. The reaction can yield products of high purity and therefore post treatment for purification may become not necessary. For these reasons, solid-state reactions are useful for commercial practicality. However, there are few studies in this area because of the limited number of such systems available[2]. Known strategies for induction of solid-state reactions include kinetic energy[3], radiation[4], and heat treatment[1,5] below the melting temperature of the target substances. Recently, a new strategy for solid-state reaction was reported for the mass production of graphene nanoplatelets via mechanochemical ball-milling[6]. Similarly, metal organic frameworks (MOFs) have also been investigated through different mechanochemical methodologies such as neat grinding[7], kneading[8], and grinding–annealing[9]. In addition, the syntheses of two-dimensional (2D) porous organic networks (PONs) from crystalline molecules were carried out by photoradiation[10–12].

On the other hand, there are numerous reports of work that utilizes different synthetic strategies to produce PONs[13] in liquid-phase reactions in the presence of suitable solvents and/or catalysts, including Sonogashira–Hagihara homo-coupling, Sonogashira–Hagihara cross-coupling, Suzuki cross-coupling, Yamamoto coupling, Friedel–Crafts alkylation, click chemistry, Gilch reaction, benzimidazole formation, Schiff-base chemistry, imidization, amidization, cyclization of three ethynyl groups, cyclization of three nitrile groups, and cyclization of three acetyl groups. The resultant PONs are expected to have high surface area, which could be useful for various applications[14–17] such as catalytic supports, gas capture and storage, energy conversion and storage, optoelectronics and semiconductors. However, the study of thermally induced solid-state explosion has not yet been reported in the field of PONs, which are expected to display different structure and properties compared to those prepared from liquid-phase processes.

Here, we introduce a synthetic methodology for the fabrication of a three-dimensional (3D) PON with high specific surface area via solid-state explosion of organic single crystals containing primer molecules. The reaction involves the Bergman reaction (cycloaromatization)[18] of 2,3,6,7,14,15-hexaethynyl-9,10-dihydro-9,10-[1,2]benzenoanthracene (HEA), which is a self-polymerizable trifunctional (M$_3$) building block with three ene-diyne groups (containing a double bond and two triple bonds). The structure of HEA single crystals is determined by single-crystal X-ray diffraction (XRD) pattern (CCDC-1475255), suggesting that two acetone and one water molecules are regularly positioned in the HEA crystal lattice. The acetone and water molecules play roles as primer to trigger explosion. As with click reactions between ethynyl and azide groups[19], thermally induced solid-state Bergman reaction of HEA crystals is completed within 0.11 s. Differential scanning calorimetry (DSC) study indicated that huge exothermic heat is explosively released during the instantaneous reaction.

## Results

**Synthesis and characterization of HEA.** The key M$_3$ building block, HEA (Supplementary Fig. 1), was prepared in three steps (see "Methods" for more details). In brief, the bromination of triptycene[20] and subsequent ethynylation yield a unique 3D HEA structure. The HEA single crystals were obtained from very slow recrystallization in a heptane/acetone mixture. The color of the as-grown bulk HEA crystals is light yellow (inset, Fig. 1a) and that of the finely ground crystal powder is off-white (inset, Fig. 1b).

The thermal behavior of HEA single crystals was studied by DSC. Typical organic single crystals display a sharp and narrow

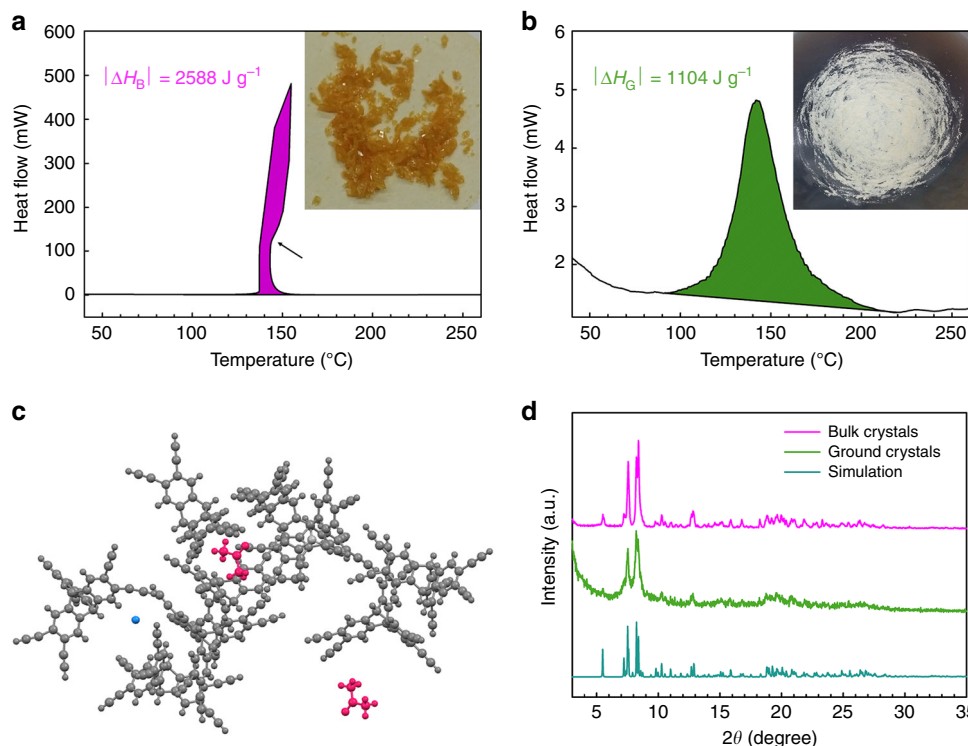

**Fig. 1** Characteristic nature of HEA crystals. DSC thermograms of samples obtained with a heating rate of 10 °C min⁻¹ under nitrogen atmosphere: **a** First heating scan of as-grown bulk HEA crystals. Inset is photograph of as-grown bulk HEA crystals; **b** First heating scan of ground HEA crystals. Inset is photograph of ground HEA crystals; **c** Ball and stick structure of crystallographic asymmetric HEA unit obtained from single-crystal X-ray diffraction (gray: HEA, red: acetone, cyan: water). **d** Experimentally determined and simulated powder XRD patterns of HEA crystals

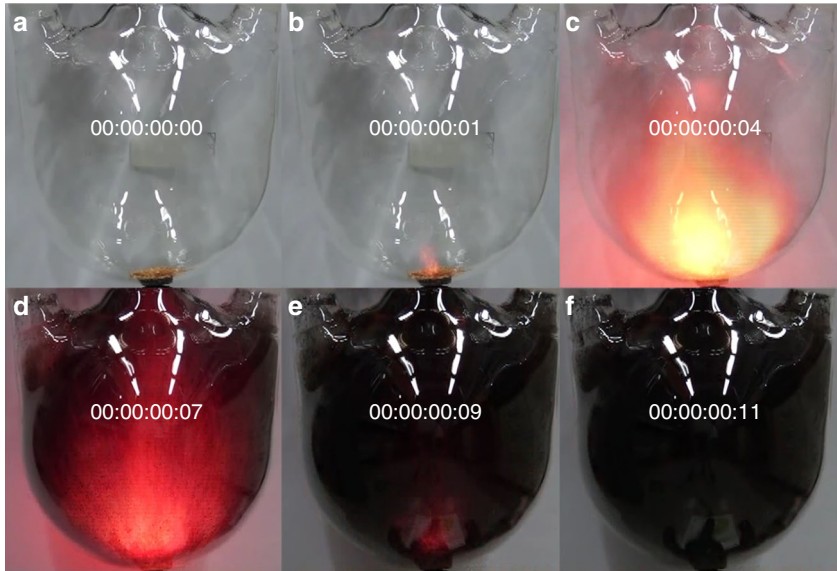

**Fig. 2** Solid-state explosion of bulk HEA crystals. **a–f** A series of photographs of explosive HEA reaction to polyHEA at different time frames. The images were captured by high-speed camera (frame 1, Supplementary Movie 1). A stepwise reaction started: **b** a minor initial ignition at 0.01 s, **d** followed by a major explosion at 0.07 s, and **f** termination at 0.11 s

endothermic crystal melting peak. For example, 1,3,5-triethynyl-benzene, which is a representative $M_3$ building block for cycloaromatization using three ethynyl groups (different from the Bergman reaction), shows an endothermic melting peak at 105.8 °C and then an exothermic reaction peak at 189.5 °C (Supplementary Fig. 2a), while ground HEA crystals only show a broad exothermic peak at 142.0 °C (Supplementary Fig. 2b). However, as-grown bulk HEA crystals display unusual thermal behavior, showing a strong exothermic peak at 137.5 °C before melting (Fig. 1a). To figure out the origin of the unusual thermal behavior, as-grown bulk HEA crystals were subjected to characterization using single-crystal XRD (see Experimental details in Supplementary Information and Supplementary Table 1). The crystallographic asymmetric unit of a single crystal consists of nine HEA molecules with two acetone and one water molecules in the lattice (Fig. 1c). The powder XRD pattern of HEA crystals was in good agreement with the simulated XRD pattern (Fig. 1d).

**Formation of polyHEA (3D-PON) and characterization**. As schematically presented in Supplementary Fig. 3, the Bergman reaction (Supplementary Fig. 3a) of enediyne groups in HEA produces a 3D PON (Supplementary Fig. 3b). Generally, the cycloaromatization occurs when enediyne groups are heated (>200 °C) in the presence of hydrogen donor in liquid phase[18,21]. Furthermore, the typical condition for cycloaromatization of three ethynyl groups (Supplementary Fig. 3c) is also a liquid phase in dried dioxane, in the presence of dicobalt octacarbonyl as catalyst[22]. In this work, however, the explosive chain reaction of HEA crystals was triggered by heat treatment in solid state (below melting temperature). Due to dramatic volume expansion during the solid-state explosion, only a small amount of HEA (0.32 g) was placed in a 4-L three-necked round-bottom flask. When the bottom of the flask where the HEA was located was rapidly heated using a heat gun, an explosive reaction was completed in 0.11 s (Fig. 2a–f and Supplementary Movie 1).

This unconventional solid-state reaction was systematically investigated by comparing the difference in calorimetric heat between as-grown bulk HEA crystals (inset, Fig. 1a, denoted as

bulk crystals) and powdered bulk HEA crystals (inset, Fig. 1b, denoted as ground crystals). After grinding the as-grown large bulk crystals into finely ground crystals, the content of acetone was significantly reduced (Supplementary Fig. 4); while that of water was increased due to the larger surface area, which caused more air moisture uptake. Although the crystallinity of the ground crystals slightly decreased, they maintained the same crystal structure as the bulk crystals (Fig. 1d). However, the reaction kinetics are dramatically different between the bulk and ground crystals upon heating (10 °C min$^{-1}$). When the primer molecules (acetone and water) in the bulk crystals accumulate enough kinetic energy, they are abruptly released (stage 1, Fig. 3a), inducing dramatic change in lattice energy (stage 2, Fig. 3a). The change in lattice energy triggers explosive Bergman reaction (stage 3, Fig. 3a). The conversion of bulk HEA crystals into PON (denoted as polyHEA) reaches almost 100%, due probably to the activation of HEA molecules by enough change in lattice energy driven by abrupt evacuation of sufficient primer molecules. On the other hand, the primer molecules in the crystal lattice could be significantly reduced in the ground crystals due to their larger surface area, which leads to less change in lattice energy. As a result, the degree of reaction in the ground crystals is lower. This primer induced stepwise reaction (a minor ignition induced by primers and a subsequent major explosion of HEA) could be repeatedly observed by a series of different experiments (Fig. 2b and Supplementary Movie 1). Such a phenomenon can be conceptually explained by reference to a small-arms cartridge. In case of a cartridge with enough primer (acetone and water molecules), all the gunpowder (HEA molecules) can be completely exploded (left inset, Fig. 3b), while a cartridge with too little primer cannot complete the explosion (right inset, Fig. 3b).

The detailed mechanism of explosive Bergman reaction is proposed as follows. When heat is applied to the bulk crystals (stage 1, Fig. 3a, b), kinetic energy is accumulated by regularly positioned acetone and water molecules in the crystal lattice. At the same time, HEA molecules in the lattice also absorb applied heat, increasing their vibration. When the combined kinetic energy exceeds the lattice energy (pink dotted line, Fig. 3a) as the temperature approaches ~140 °C (much lower than thermal

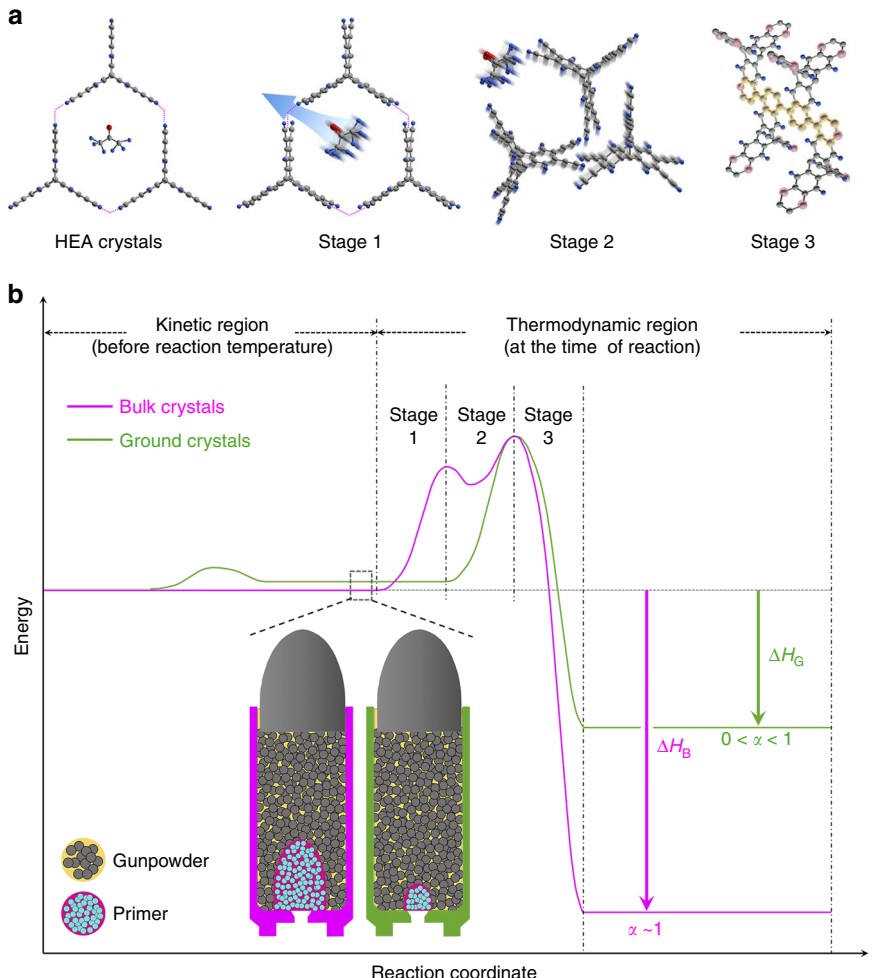

**Fig. 3** Proposed mechanism and thermodynamics of solid-state Bergman reaction. **a** Schematic representation of the explosive transformation from HEA crystals to polyHEA (gray: carbon, red: oxygen, cyan: hydrogen). The pink dotted lines represent lattice energy of HEA crystals. **b** Energy diagrams of the bulk and ground crystals based on exothermic reaction heats from DSC measurements (Fig. 1a, b) and DFT calculations (Supplementary Figs. 5 and 6). Insets represent small-arms cartridges, explaining the reaction conditions for the bulk crystals (left: containing more primer molecules) and the ground crystals (right: containing less primer molecules) shortly before explosion. $\Delta H_B$ and $\Delta H_G$ stand for reaction enthalpies of the bulk and ground crystals, respectively

cyclization of enediyne unit, >200 °C), HEA molecules in the crystal lattice rearrange into a less stable state (stage 2, Fig. 3a, b) upon abrupt release of acetone and water molecules (an initial minor ignition, Supplementary Movie 1). At the same time, enediyne groups are close enough for cycloaromatization (forming extended aromatic phenyl rings, stage 3, Fig. 3a, b), which generates tremendous reaction enthalpy ($\Delta H_B$) and further accelerates the cyclization of all the enediyne groups of HEA into polyHEA, ($\alpha \sim 1$, $\alpha$: degree of reaction conversion, a subsequent major explosion, Supplementary Movie 1). During the explosive reaction, there is a dramatic temperature decrease from 151 to 143 °C (arrow, Fig. 1a), which indicates a huge volume expansion due to the ejection of primer molecules that evacuate substantial heat from the system (sample pan). On the other hand, the ground crystals containing the less primer molecules release the less lattice energy ($0 < \alpha < 1$, insufficient to complete the explosive reaction of HEA crystals into polyHEA). The indications in the proposed energy diagram (Fig. 3b) were also supported by density functional theory (DFT) calculations (see Supplementary Figs. 5 and 6 and Supplementary Note 1, 2).

On the basis of these results, the released energy induced by cycloaromatization of enediyne groups in the ground crystals is expected to be much less than that in the bulk crystals. The

tremendous difference in specific calorimetric heats between the bulk (2588 J g$^{-1}$, Fig. 1a) and ground crystals (1104 J g$^{-1}$, Fig. 1b) could be associated with the difference in overall reaction conversion ($\alpha$). In order to understand the relationship between the change in lattice energy and reaction kinetics, control experiments were carried out with respect to the heating rate (Supplementary Figs. 7–10 and Supplementary Note 3–6). The results agreed well with the hypothesis that the explosion is associated with change in lattice energy per given time.

The formation of polyHEA structure was characterized using Fourier transform infrared (FT-IR) spectroscopy (Supplementary Fig. 11a). In the case of as-grown HEA crystals, the band at 3300 cm$^{-1}$ could be assigned to $sp$ C–H stretching from ethynyl units on enediyne groups. This peak disappeared after explosion, suggesting that the reaction was completed without an unreacted enediyne moiety. As shown in a magnified spectrum (inset, Supplementary Fig. 11a), the bands at 2970 and 2927 cm$^{-1}$ are due to the stretching of $sp^2$ C–H and $sp^3$ C–H, respectively. The peaks, centered at 2921 and 2857 cm$^{-1}$ after explosion, are characteristic of triptycene units in polyHEA. To further investigate the chemical structure of the resultant polyHEA, solid-state magic-angle carbon thirteen nuclear magnetic resonance ($^{13}$C NMR) spectroscopy was utilized. The solid $^{13}$C NMR

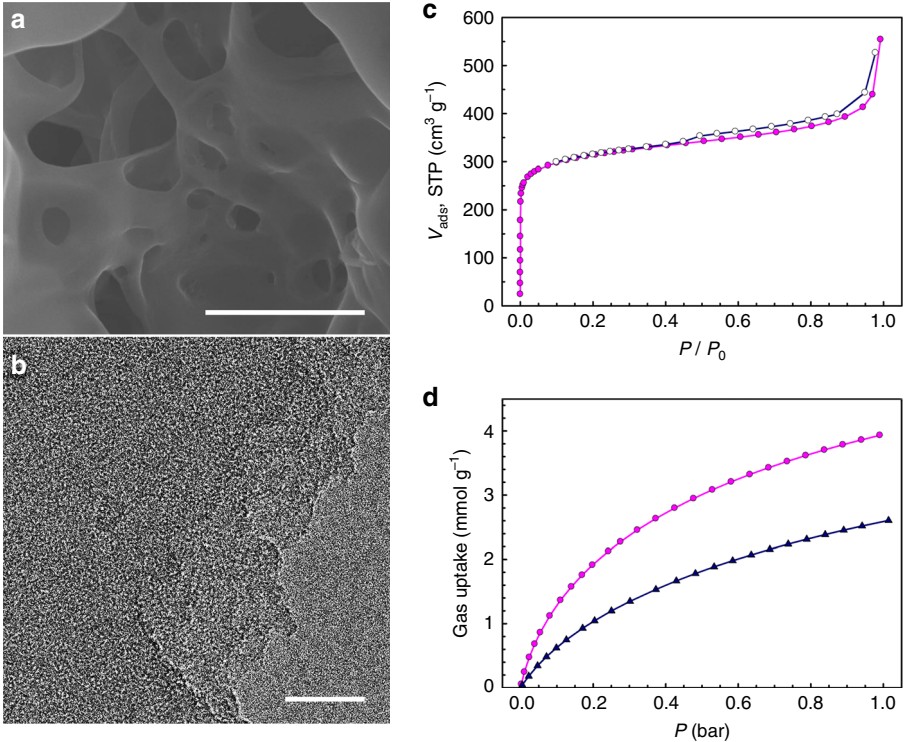

**Fig. 4** Morphology and gas sorption properties of polyHEA. **a** SEM image of polyHEA (scale bar is 2 µm). **b** TEM image of polyHEA (scale bar is 50 nm). **c** Nitrogen adsorption (solid circle) and desorption (open circle) isotherms of polyHEA at 77 K. **d** $CO_2$ adsorption isotherms of polyHEA at 273 K (pink circle) and 298 K (dark blue triangle)

spectrum of the polyHEA in Supplementary Fig. 11b shows a single broad peak at 137.4 ppm, which is associated with aromatic carbon.

The X-ray photoelectron spectroscopy spectrum shows only C 1s and O 1s peaks (Supplementary Fig. 11c). High-resolution survey spectrum of C 1s indicated a single peak at 284.3 eV ($sp^2$ C–C) (Supplementary Fig. 11c), which agrees well with the $^{13}C$ NMR result (Supplementary Fig. 11b). In addition, polyHEA contains high carbon content (96.75 at%) and low oxygen content (3.25 at%). The O 1s peak can be deconvoluted into two peaks at 533.1 eV (C-OH) and 532.0 eV (C=O), associated mostly with physically absorbed moisture (Supplementary Fig. 11e)[23]. Energy dispersive X-ray spectroscopy using field-emission scanning electron microscopy (FE-SEM) also detected predominantly carbon (96.61 at%) and little oxygen (3.39 at%) (Supplementary Fig. 12). The overall elemental composition of polyHEA is summarized in Supplementary Table 2.

As with most of the 3D PONs that are known to be amorphous solids[24], the powder XRD pattern of polyHEA is featureless, suggesting an amorphous nature for polyHEA (Supplementary Fig. 13a). Thermogravimetric analysis indicates that polyHEA is thermally stable up to 450 °C in air, and its char yield at 1000 °C is over 90 wt% in nitrogen (Supplementary Fig. 13b). FE-SEM and high-resolution transmission electron microscopy were utilized to investigate the morphology of polyHEA. SEM images (Fig. 4a) show the formation of fishing-net-type morphology, which is different from those of previously reported 3D PONs with spherical morphology prepared from liquid-phase reaction[25,26]. By changing the reaction conditions (heating rates), the dimension of the macroscopic pore could be controlled (Supplementary Fig. 14). The high-resolution transmission electron microscopy image shows uniform pores (dark and bright spots) in the polyHEA matrix (Fig. 4b), suggesting the porous nature of polyHEA.

**BET surface area and carbon dioxide adsorption study of polyHEA.** In order to figure out the porosity and gas sorption properties, nitrogen gas adsorption/desorption isotherm was tested at 77 K (Fig. 4c). The specific Brunauer–Emmet–Teller (BET) surface area ($S_{BET}$) and total pore volume are 1176 $m^2 g^{-1}$ and 0.843 $cm^3 g^{-1}$, respectively. The polyHEA exhibited a Type-1 isotherm with microporous (pore size<2 nm) nature[27]. The apparent hysteresis around 0.45 $P/P_0$ between adsorption/desorption was observed, indicating mesoporous (pore size 2–50 nm) materials was attributed to pore network effects[28]. The pore size distribution for polyHEA as calculated using the Grand Canonical Monte Carlo method. The widths of the two major pore types of polyHEA were 0.89 and 3.43 nm (Supplementary Fig. 15a).

As shown in Fig. 4d and Supplementary Fig. 15b, the carbon dioxide ($CO_2$) isotherms were collected and their isosteric heats of adsorption ($Q_{st}$) were also calculated using the Clausius Clapeyron equation. The polyHEA exhibited $CO_2$ uptake of 3.93 mmol $g^{-1}$ at 273 K and 2.61 mmol $g^{-1}$ at 298 K (1 bar). The $Q_{st}$ of polyHEA for $CO_2$ was found to be 31.7 kJ $mol^{-1}$ at zero coverage. $CO_2$ uptake is higher than for reported PONs having very high surface area, such as PAF-1 ($S_{BET}$: 5460 $m^2 g^{-1}$, 2.09 mmol $g^{-1}$ at 273 K and 1 bar)[29], Network-A ($S_{BET}$: 4077 $m^2 g^{-1}$, 2.65 mmol $g^{-1}$ at 273 K and 1 bar)[30] and BPL carbon ($S_{BET}$: 1150 $m^2 g^{-1}$, 2.09 mmol $g^{-1}$ at 273 K and 1 bar; a common reference material for $CO_2$ uptake)[30]. For determining the $CO_2$ capture capacity, the surface properties and tuned pore geometry of porous materials is more important than large surface area[31].

In principle, the PONs with nitrogen-rich functionalities, such as triazine, tetrazole, imidazole, carbazole, phosphazene, imide, amine, and azo compounds exhibit high $CO_2$ adsorption capacities. This is because of the strong electrostatic interactions between $CO_2$ and nitrogen sites[26]. However, $CO_2$ uptake and $Q_{st}$

value of polyHEA without nitrogen are comparable to those of nitrogen-containing PONs, such as SNW-1 (3.64 mmol g$^{-1}$ at 273 K and 1 bar, $Q_{st}$: 35.0 kJ mol$^{-1}$)[32], PECONF-3 (3.49 mmol g$^{-1}$ at 273 K and 1 bar, $Q_{st}$: 26.0 kJ mol$^{-1}$)[25], and azo-COP-2 (2.55 mmol g$^{-1}$ at 273 K and 1 bar, $Q_{st}$: 24.8 kJ mol$^{-1}$)[33]. This is because electron rich cavities in polyHEA interact with the carbon atoms of $CO_2$ molecules. The porous properties of PONs with and without nitrogen are compared in Supplementary Table 3.

## Discussion

We were able to synthesize an M$_3$ building block, HEA, and to form its large single crystals by slow solvent evaporation. The structure of HEA crystals by single-crystal XRD analysis revealed the presence of primer molecules (acetone and water) in the HEA crystal lattice. The evacuation of primer molecules could induce abrupt change in lattice energy, which triggers explosive Bergman reaction (cycloaromatization) of HEA crystals into 3D-network-structured polyHEA in solid-state (~140 °C, below melting temperature), without the presence of solvent(s) and catalyst(s). The resultant polyHEA is a porous material with high specific surface area ($S_{BET}$: 1176 m$^2$ g$^{-1}$) and displays unusual sorption capacity of carbon dioxide ($CO_2$). This new synthetic protocol may pave the way to design and synthesize molecules suitable for the solid-state formation of other PONs for various applications beyond those produced in liquid-phase processes.

## Methods

Detailed information on materials and instrumentations is provided in Supplementary Information.

**Synthesis of 2,3,6,7,14,15-hexabromo-9,10-dihydro-9,10-[1,2]benzenoanthracene (HBA).** HBA was prepared according to literature[34].

**Synthesis of 2,3,6,7,14,15-hexakis((trimethylsilyl)ethynyl)-9,10-dihydro-9,10-[1,2]benzenoanthracene (HMSA).** 2,3,6,7,14,15-Hexabromo-9,10-dihydro-9,10-[1,2]benzenoanthracene (2.0 g, 2.748 mmol), CuI (0.0312 g, 0.164 mmol), PdCl$_2$(PPh$_3$)$_2$ (0.1732 g, 0.247 mmol), and PPh$_3$ (0.1302 g, 0.496 mmol) were placed in a 250 mL round-bottom flask. Anhydrous i-Pr$_2$NH (150 mL) and tri-methylsilylacetylene (4 mL, 28.1 mmol) were added in the flask. The reaction mixture was heated and refluxed under nitrogen atmosphere for 24 h and then allowed to cool to room temperature. The reaction mixture was filtered through a pad of Celite and washed with diethyl ether. Filtrate was dried in rotary evaporator, and CH$_2$Cl$_2$ was added to dissolve the residue and filtered again through a pad of silica gel to remove metallic impurities. The filtrate was concentrated on a rotary evaporator under reduced pressure and loaded to a silica gel column. Hexane/ethyl acetate (100/5, v/v) was used as an eluent. Upon removal of solvent, yellow gel-type product was collected (1.862 g, 81.7% yield). $^1$H NMR (400 MHz, CDCl$_3$, $\delta$ = ppm): 0.229 (26.71, Si–CH$_3$), 5.204 (1.00, CH), and 7.396 (2.90, Ar–H). $^{13}$C NMR (400 MHz, CDCl$_3$, $\delta$ = ppm): 0.146, 52.65, 98.38, 103.3, 123.6, 127.6, and 143.4.

**Synthesis of 2,3,6,7,14,15-hexaethynyl-9,10-dihydro-9,10-[1,2]benzenoanthracene (HEA).** In a 250 mL round-bottom flask, a solution of NaOH (0.7 g, 17.5 mmol) in methanol (20 mL) was placed and a solution of 2,3,6,7,14,15-hexakis ((trimethylsilyl)ethynyl)-9,10-dihydro-9,10-[1,2]benzenoanthracene (1.0 g, 1.20 mmol) in CH$_2$Cl$_2$ (20 mL) was slowly added. The mixture was stirred at room temperature for 12 h. The reaction mixture was dried on a rotary evaporator under reduced pressure and CH$_2$Cl$_2$ (30 mL) was added to dissolve the residue and the white insoluble solids were filtered off. The filtrate was then washed with brine (20 mL) and the organic phase was extracted with CH$_2$Cl$_2$ using a separation funnel. The separated organic phase was dried over anhydrous Na$_2$SO$_4$ and filtered. The filtrate was concentrated on a rotary evaporator under reduced pressure and loaded to a silica gel column. Hexane/ethyl acetate (10/1, v/v) was used as an eluent, off-white solid powder (0.349 g, 72.9%) was obtained after removal of the solvent. Large HEA single crystals was grown in acetone/heptane mixture with a very slow evaporation of solvents at room temperature. $^1$H NMR (400 MHz, CDCl$_3$, $\delta$ = ppm): 3.266 (2.39, ≡C–H), 5.330 (1.00, C–H), and 7.502 (2.69, Ar–H). $^{13}$C NMR (400 MHz, CDCl$_3$, $\delta$ = ppm): 52.51, 82.37, 82.87, 123.6, 129.0, and 145.41.

**Synthesis of polyHEA.** Caution: This reaction is extremely exothermic and proceeds with violent explosion; due careful attention is necessary. As-grown bulk HEA crystals (0.32 g) was placed in a 4-L reaction flask under reduced pressure.

The HEA crystals placed at the bottom of flask, where was rapidly heated by heat gun. After explosive reaction, very light-weight floating products in the flask were rinsed with acetone and collected by suction filtration. The product was Soxhlet extracted with acetone for 2 days to get rid of unreacted HEA and then with water to remove other water-soluble impurities for 2 days, if any. Finally, the sample was freeze-dried under reduced pressure (0.5 mmHg) at –120 °C for 2 days to give 0.30 g (94% yield, considering loss during collection and work-up procedures, it is essentially quantitative yield) of black powder.

**Data availability**. Supplementary information, materials characterizations and supplementary movie are available in the online version of the paper.  Correspondence and requests for materials should be addressed to J.-B.B.

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

## Acknowledgements

This work was supported by the Creative Research Initiative (CRI, 2014R1A3A2069102), Science Research Center (SRC, 2016R1A5A1009405), Climate Change (2016M1A2940910) and BK21 Plus (10Z20130011057) programs through the National Research Foundation (NRF) of Korea. N.P. and D.S. were supported by NRF-2017R1A4A1015323.

## Author contributions

J.-B.B. conceived the HEA superstructure and oversaw all the research phases. J.-B.B., S.-Y.B. and J.M. designed, synthesized and characterized the samples. D.K. and M.S.L. carried out the crystallographic study of HEA crystals. D.S. and N.P. were involved in the DFT study of the new material. S.-M.J. carried out the TEM study. J.M., I.-Y.J., S.-J.K. and S.-H.S. interpreted the experimental data and designed the schematic representation of reaction. J.-B.B., S.-Y.B. and J.M. wrote the paper and discussed the results. All the authors contributed to and commented on this manuscript.

## Additional information

**Competing interests:** The authors declare no competing financial interests.

