## [Peer Review File · Nature Communications]

Reviewers' comments:

Reviewer #1 (Remarks to the Author):

This paper is a delight! There are many researchers working in the area of porous materials and it is quite rare that something completely different comes along - but this paper fits that category. The practical utility of this method is unclear, and the porosity of the resulting material is fairly unremarkable, but I would strongly support publication in Nature Communications on grounds of novelty - and also because this could have wide breadth of application, extending perhaps beyond this alkyne chemistry.

A few comments:

(1) This paper surely warrants a Safety Note in the Methods section. The supporting video (which is fantastic!) shows some precautions (e.g., an expansion tube) that are not alluded to in the section entitled "Synthesis of polyHEA". I would also suggest that this reaction ought not to be scaled up without adequate precautions - may sound obvious, but I think it is the duty of the authors and the journal to point this out.

(2) Is it clear that the primer molecules are needed? I guess it is hard to sublime this compound (!!) but is it not possible that the pure, unsolvated HEA molecule is inherently explosive? I am not surprised that the molecule forms a solvate (see papers referred to below in (3)), but if the solvent can be removed (by evaluation or careful solvent exchange) then it would be possible to test the need for the "primer". (I inferred from the introduction - top of page 6 - that the melting behaviour was different in the solvate, but this point could do with some elaboration).

(3) The structure of HEA is closely related to certain intrinsically porous crystals and the following two papers in particular should be cited:

M. Mastalerz & I. M. Oppel, *Angew. Chem. Int. Ed.*, 2012, 51, 5252.

A. Pulido et al., *Nature*, 2017, 543, 657.

(Indeed, the structural motif in Fig 3a looks isostructural with one of the phases in the 2017 Nature paper.)

In summary - a great little paper, quite unlike anything else I have seen - I strongly support publication here.

Reviewer #2 (Remarks to the Author):

The paper submitted by Baek et al shows that a three-dimensional porous organic network (PON) can be obtained by using the solid-state explosion of organic single crystals. Even if there are numerous reports of work that utilizes different synthetic strategies to produce PONs, this study reveals a new synthetic methodology (the thermally induced solid-state explosion) for the fabrication of a three-dimensional PON that shows high specific surface area and good sorption capacity of carbon dioxide (CO₂). The paper is well organized, easy to read. The data are convincing and support the discussion. I recommend the publication of this paper in Nature Communications after a minor revision.

(1) About the nitrogen gas adsorption/desorption isotherms, the authors claimed that "the polyHEA

exhibited a Type-1 isotherm with microporous (pore size < 2 nm) nature. The apparent hysteresis around 0.45 P/P0 between adsorption/desorption was observed, indicating mesoporous (pore size 2–50 nm) materials was attributed to pore network effects.” If there is the presence of the hysteresis between adsorption/desorption, it should be a Type-IV isotherm with mesoporous nature but not a microporous nature.

(2) The caption in the Supplementary Figure 5 is different with its figure.

(3) As shown in Supplementary Figure 6, the formation energy is -189.48 KJ mol⁻¹ for polyHEA in the figure; however, it is -1250.57 KJ mol⁻¹ in the caption. Which one is the right number?

Reviewer #3 (Remarks to the Author):

I like the idea! But the product is some amorphous, cross-linked carbon, and doesn't seem to be anything special beyond carbon. Maybe try to control the reaction a bit more? Maybe suspend the crystals in water and irradiate? Or heat - sealed tube in water maybe? Of course, be careful with that - just a few crystals.

Response to the Reviewers

Jong-Beom Baek

To begin, we would like to thank the referees for their time spent on reviewing our manuscript and giving us their thoughtful and constructive comments. As this is a submission to Nature Communication, the Reviewers' remarks have been very carefully considered and we tried to clarify their concerns.

Reviewers' comments:

Reviewer #1 (Remarks to the Author):

This paper is a delight! There are many researchers working in the area of porous materials and it is quite rare that something completely different comes along - but this paper fits that category. The practical utility of this method is unclear, and the porosity of the resulting material is fairly unremarkable, but I would strongly support publication in Nature Communications on grounds of novelty - and also because this could have wide breadth of application, extending perhaps beyond this alkyne chemistry.

A few comments:

Comment 1-1: This paper surely warrants a Safety Note in the Methods section. The supporting video (which is fantastic!) shows some precautions (e.g., an expansion tube) that are not alluded to in the section entitled "Synthesis of polyHEA". I would also suggest that this reaction ought not to be scaled up without adequate precautions - may sound obvious, but I think it is the duty of the authors and the journal to point this out.

Response 1-1: We truly appreciate the reviewer's very important comment on safety issue. In response to the reviewer's suggestion, we have added the suggested precautionary sentence in the synthesis part (Experimental Section). Once again, million thanks!

Comment 1-2: Is it clear that the primer molecules are needed? I guess it is hard to sublime this compound (!!) but is it not possible that the pure, unsolvated HEA molecule is inherently explosive? I am not surprised that the molecule forms a solvate (see papers referred to below in (3)), but if the solvent can be removed (by evaluation or careful solvent exchange) then it would be possible to test the need for the "primer". (I inferred from the introduction - top of page 6 - that the melting behaviour was different in the solvate, but this point could do with some elaboration).

Response 1-2: Many thanks for the suggestion. Unfortunately, however, experiments using pure HEA crystal (without guest primer molecules) are quite difficult. It is because highly

active ethynyl groups (particularly, in enediyne unit), which tend to start reaction in common organic solvents with heating. To exclude primer molecules (solvents) and thus to prepare pure HEA crystal by recrystallization with various organic solvents (CH_2Cl_2 , acetone with methanol/ethanol/ethyl acetate/hexane/heptane) were also tried. However, only large yellow crystals (bulk HEA crystal) could only be obtained from two solvent system (acetone and heptane mixture) by utilizing the big difference in evaporation rate at room temperature without applying heat. Thus, at current moment, it is not quite possible to conclude the roles of other solvents, which can also trigger the Bergman reaction of HEA.

In brief, due to the sensitivity of multiple enediyne groups, the difficulties to perfectly remove organic solvents in HEA crystal. However, the control experiment clearly indicates that HEA crystal with less primer molecules cannot complete the explosive reaction (Supplementary Figure 7).

Hence, as like the formation of polyHEA, it is very likely that new enediyne-containing molecules with other guest molecules, such as organic solvents, can also be possible for the primer-induced solid-state Bergman reaction, and thus this system could suggest a new trend setting research field in the future.

Comment 1-3: The structure of HEA is closely related to certain intrinsically porous crystals and the following two papers in particular should be cited:

M. Mastalerz & I. M. Oppel, *Angew. Chem. Int. Ed.*, 2012, 51, 5252.

A. Pulido et al., *Nature*, 2017, 543, 657.

(Indeed, the structural motif in Fig 3a looks isostructural with one of the phases in the 2017 *Nature* paper.)

In summary - a great little paper, quite unlike anything else I have seen - I strongly support publication here.

Response 1-3: We highly appreciated the reviewer for the suggestion. The two suggested references have been added to improve the literature coverage of the manuscript.

Reviewer #2 (Remarks to the Author):

The paper submitted by Baek et al shows that a three-dimensional porous organic network (PON) can be obtained by using the solid-state explosion of organic single crystals. Even if there are numerous reports of work that utilizes different synthetic strategies to produce PONs, this study reveals a new synthetic methodology (the thermally induced solid-state explosion) for the fabrication of a three-dimensional PON that shows high specific surface area and good sorption capacity of carbon dioxide (CO_2). The paper is well organized, easy to read. The data are convincing and support the discussion. I recommend the publication of this paper in *Nature Communications* after a minor revision.

Comment 2-1: About the nitrogen gas adsorption/desorption isotherms, the authors claimed that “the polyHEA exhibited a Type-1 isotherm with microporous (pore size < 2 nm) nature. The apparent hysteresis around 0.45 P/P0 between adsorption/desorption was observed, indicating mesoporous (pore size 2–50 nm) materials was attributed to pore network effects.” If there is the presence of the hysteresis between adsorption/desorption, it should be a Type-IV isotherm with mesoporous nature but not a microporous nature.

Response 2-1: Thanks for the comment and letting us double check the type of isotherms from the IUPAC system. Actually, the isotherm is indeed type-I instead of type-IV, but the hysteresis in desorption curve is related to H4 type, which is related to the type of hysteresis. In H4 type hysteresis, the pronounced uptake at low p/p0 is related to the filling of micropores. The H4 type hysteresis is often found in microporous carbon structure (Thommes Matthias *et al. Pure & Applied Chemistry* **2015**, 87, 1051-1069).

Comment 2-2: The caption in the Supplementary Figure 5 is different with its figure.

Response 2-2: Thanks to reviewer for pointing this important mistake. The manuscript has updated with corrected Figure 5 caption. In this figure, it is emphasized that the Bergman reaction can be energetically stable by chemical bonding between sp^2 carbon radicals.

Comment 2-3: As shown in Supplementary Figure 6, the formation energy is -189.48 KJ mol⁻¹ for polyHEA in the figure; however, it is -1250.57 KJ mol⁻¹ in the caption. Which one is the right number?

Response 2-3: We are very sorry for creating confusion again. Formation energy of -1250.57 KJ mol⁻¹ is correct value for polyHEA. This huge formation energy gives reasonable evidence for stabilization of polyHEA after the Bergman reaction.

Reviewer #3 (Remarks to the Author):

Comment 3-1: I like the idea! But the product is some amorphous, cross-linked carbon, and doesn't seem to be anything special beyond carbon. Maybe try to control the reaction a bit more? Maybe suspend the crystals in water and irradiate? Or heat - sealed tube in water maybe? Of course, be careful with that - just a few crystals.

Response 3-1. As a matter of fact, we tried to polymerize the single crystal by irradiation using UV light, X-ray and gamma ray. Despite enough irradiation time, the ethynyl groups in HEA crystals remained unchanged (chemical structure of the HEA crystals, progress of the reaction, before and after irradiation was monitored using NMR). We have come up with the conclusion that the intermolecular distance between the ethynyl groups in HEA molecules is too far to contact for the reaction (Fig. 1c). Possibly these wavelengths (UV, X-ray, and gamma ray) could not evaporate acetone and water (primer molecules) to demolish HEA crystals to bring the ethynyl groups closer enough for intermolecular reaction; as the distance

between the ethynyl groups is too far to cause the reaction to happen without removal of acetone and water (primer molecules).

Carrying out the reaction in water is a very interesting idea, but we feel water reaction may have limitations at ambient pressure unless using autoclave as the reviewer suggested. First, based on DSC result from as-grown bulk HEA crystal, which shows a strong exothermic peak at 137 °C. Thus, increasing the temperature, at which the reaction can occur, requires autoclave with extreme caution. Second, as we explained with control experiments (Supplementary Figure 7), the heating rate of bulk HEA crystal is very important factor to induce complete explosive reaction. If we conduct the reaction in water, heating rate will be one of limiting factors. Thus, we are afraid of the less feasibility of experiments.

Nevertheless, we do appreciate this very interesting suggestion. We are going to further study the possibility with this point in mind and the results will be reported in the future..